# Nanopriming with Au-NPs Enhances Tomato Metabolism and Cold Tolerance

**DOI:** 10.3390/plants15010083

**Published:** 2025-12-26

**Authors:** Yuliya Venzhik, Natalia Naraikina, Alexander Deryabin, Kseniya Zhukova, Mariya Snigur, Alexander Kartashov, Ivan Kochetkov, Lev Dykman

**Affiliations:** 1K.A. Timiryazev Institute of Plant Physiology, Russian Academy of Sciences, 127276 Moscow, Russia; naraikina@ifr.moscow (N.N.); anderyabin@ifr.moscow (A.D.); zhukova@ifr.moscow (K.Z.); snigur@ifr.moscow (M.S.); botanius@yandex.ru (A.K.); kochetkov@ifr.moscow (I.K.); 2Institute of Biochemistry and Physiology of Plants and Microorganisms, Saratov Scientific Centre of the Russian Academy of Sciences, 410049 Saratov, Russia; dykman_l@ibppm.ru

**Keywords:** *Solanum lycopersicum*, nanopriming, gold nanoparticles, photosynthetic apparatus, pro-/antioxidant balance, antioxidant system, cold tolerance

## Abstract

Nanopriming is a unique way to change the metabolism and stress tolerance of plants. In this study, the effects of seed priming with gold nanoparticles (Au-NPs, 15 nm) on the photosynthetic apparatus, pro-/antioxidant balance, and cold tolerance of tomato (*Solanum lycopersicum* L.) were examined for the first time. The study revealed that nanopriming with Au-NPs (20 µg mL^−1^ for 24 h) causes an increase in cold tolerance both under control conditions and after cold adaptation at 9 °C for 5 d. In additional, nanopriming leads to the formation of tomato plants with altered photosynthetic apparatus and pro-/antioxidant status. At the same time, nanoparticles do not accumulate in the aboveground parts of the plants. Under control conditions, nanopriming enhances photosynthesis rate, the relative expression of *rbcS* gene, the number of grana in chloroplast, and the content of photosynthetic pigments in leaves. Regardless of the temperature (in control conditions and during cold adaptation), nanopriming increases the size of chloroplast, the relative expression of *rbcL* gene, the content of malonic dialdehyde and H_2_O_2_, and the activity of catalase and superoxide dismutase in leaves. Nanopriming decreases the activity of ascorbate peroxidase and sugar content in tomato leaves and leads to a number of structural changes in chloroplasts. This study highlights the potential of the use of Au-NPs as a promising strategy to mitigate cold stress in tomato plants by affecting photosynthesis and the antioxidant system.

## 1. Introduction

The demand for food commodities has led to an acute shortage of food resources in countries facing agricultural problems. Human pressures on natural communities and climate instability aggravate this issue. Lots of agricultural plants suffer from low temperature (LT) every year [1]. Prolonged action of LT retards plant growth, photosynthesis, respiration, and water and mineral metabolism, causing oxidative stress [2,3]. Therefore, the search for new methods to increase plant tolerance to LT is highly relevant. Nanoparticles (NPs) have special chemical and physical characteristics and can influence plant metabolism and stress tolerance [4,5]. The possible mechanisms of this process are extensively discussed in the literature. There is evidence that some NPs are able to directly regulate stress response genes. ZnO-NPs affect the expression of transcription factors (*ZIP*, *MIB*, and *WRKY*) involved in the response to cold stress [6] and increase the expression of the *hsfa2*, *hsp70*, and *hsp101* genes, which are involved in the production of heat shock proteins [7]. Ag-NPs alter the expression of genes involved in the stress response to salt [8] and LT stress [9]. Studies show that NPs act as stress signaling molecules that “switch on” the molecular mechanisms of adaptation processes and trigger classical stress signaling reactions, such as the Ca^2+^-associated signaling pathway. As shown in studies using a proteomic approach, Ag-NPs act on proteins associated with calcium signaling [10]. The authors hypothesized that Ag+ released from the surface of NPs enters into cellular metabolism by binding to Ca^2+^ receptors and Ca^2+/^Na^+^ channels. It is assumed that NPs can replace Ca^2+^ in channels and imitate the action of Ca^2+^, causing a cascade of stress reactions. This assumption is supported by the fact that CuO-NPs, precisely through Ca^2+^ signaling, trigger salt-sensitive (SOS) signaling pathways [11].

Gold nanoparticles (Au-NPs) have several advantages over other types of NPs [12]. On the one hand, Au-NPs can be produced fast and easily using low-cost chemical reagents. Aqueous colloidal solutions of Au-NPs are synthesized from chloroauric acid and sodium citrate [13]. The obtained solutions contain Au-NPs with high stability and biocompatibility [14]. On the other hand, Au-NPs in microdoses up to 100 μg mL^−1^ are often non-toxic to plants, animals, and people. Therefore, Au-NPs are widely used for biomedical purposes and are considered safe for plants, animals, and humans [15,16,17]. Au-NPs with an average diameter of 5–50 nm most often have a stimulatory effect on plants [18]. The presence of unique optical properties associated with the excitation of localized plasmon resonances during light interaction is responsible for the worldwide scientific interest in the use of Au-NPs in biology [13,14]. In this context, Au-NPs have garnered much interest recently as compounds that may improve crop environmental tolerance. Importantly, research on the impact of Au-NPs on plant stress tolerance is limited. Research on barley and *Arabidopsis* has demonstrated that applying Au-NPs to roots causes an increase in cell wall stiffness and changes the expression of the *DGR1* and *DGR2* genes, which encode cell wall proteins [18,19]. The results showed that the Au-NPs control the expression of genes encoding proteolytic enzymes and anion carrier proteins. Therefore, Au-NPs can integrate into the plasmalemma and activate cellulose synthases and anhydrases, controlling proteolytic activity [9]. Au-NPs limit Cd uptake by the root cells of rice plants [20,21] and increase the activity of antioxidant system (AOS) enzymes under saline conditions [22]. In this sense, the possibility of further incorporating Au-NPs into agriculture and biology is promising. Research on these compounds that increase plant tolerance to LT is crucial, particularly for cold-sensitive plants with limited adaptive potential.

In recent years, nanopriming, i.e., soaking seeds in NP solutions at low concentrations for a certain time, has gained popularity [23,24,25]. The use of this method allows the production of plants whose metabolism is altered. Moreover, NPs and their constituent metal ions are absent or present in trace amounts in the aboveground parts of plants growing from treated seeds [23]. Nanopriming is considered to be the most efficient, economical, and environmentally friendly strategy for utilizing NPs in plants [24,26].

We assume that nanopriming would change the metabolism of plants not only under control conditions but also during cold adaptation (CA). The object of this investigation is tomato (*Solanum lycopersicum* L.), which is a cold-sensitive plant. Tomato plants can adapt to LTs in the range of 6 °C to 9 °C, but temperatures below this limit are considered disastrous. The effects of nanopriming with colloidal Au-NPs solution (15 nm, 20 µg mL^−1^) on cold tolerance, the ultrastructure of mesophyll cells, the activity of the photosynthetic apparatus (PSA), and some parameters related to the pro-/antioxidant balance in tomato plants under control conditions (at 22 °C) and after cold adaptation (CA) at 9 °C for 5 d were studied.

## 2. Results

### 2.1. Nanopriming in Control Conditions (At 22 °C)

The experiments revealed that nanopriming in control conditions did not affect the shoot height of tomato plants (Table 1). However, a stimulatory effect of nanopriming on the accumulation of shoot dry weight (DW) was observed (Table 1).

In addition, nanopriming changed the ultrastructure of chloroplasts in leaves of tomato; in particular, the chloroplast size and number of grana and starch inclusions were increased under the influence of Au-NPs (Figure 1a,b; Table 2 and Table 3). However, the number of plastoglobuli decreased after nanopriming. Au-NPs did not affect the area of grana, plastoglobuli, and starch inclusions in chloroplasts (Table 3) or the size of mitochondria and peroxisomes in leaf cells of tomato (Table 2).

At the same time, nanopriming increased the photosynthetic rate by 30% but did not affect the respiration rate (Figure 2). Moreover, in the plants treated with the Au-NPs, the relative expression levels of the genes encoding the large (*rbcL*) and small (*rbcS*) subunits of ribulose-1,5-bisphosphate carboxylase/oxygenase (RuBisCo) were increased (Figure 3). Nanopriming also increased the content of chlorophylls and carotenoids in tomato leaves (Table 1).

The results of the experiments revealed that the contents of malondialdehyde (MDA) and H_2_O_2_ in the leaves of Au-NPs-treated plants were almost twice as high as those in the leaves of the control plants (Table 4). In addition, the activities of antioxidant enzymes such as catalase (CAT), guaiacol peroxidase (POX), and superoxide dismutase (SOD) were several times greater in the plants of the nanopriming variant. However, the activity of ascorbate peroxidase (APX) was noticeably lower in plants treated with Au-NPs (Table 4). Au-NPs treatment also slightly reduced the content of soluble sugars (glucose, fructose, and sucrose) in tomato leaves (Table 1).

The leakage of electrolytes from the tissues of tomato leaves was analyzed as an indicator of tolerance to cold (Table 5). Damaging temperatures (0–4 °C) were used in this experiment. The effect of nanopriming on electrolyte leakage was clearly visible in the plants cooled to 4 °C; nanopriming reduced electrolyte leakage by 5% (Table 5). At 2 °C and 0 °C, all the plants (irrespective of the NPs treatment) died: the amount of electrolyte leakage from their tissues was greater than 80% (Table 5).

### 2.2. Nanopriming During Cold Adaptation (9 °C)

The experiments revealed that the plants continued to grow during CA, but nanopriming did not affect this process. Nanopriming stimulated the accumulation of shoot DW during CA (Table 1).

In additional, nanopriming led to the formation of rounded chloroplasts, in which the membrane system shifted to the periphery (Figure 1d). Large, dark peroxisomes (Figure 4b,c); clusters of organelles (Figure 4a,c); and invaginations of the cytoplasm within the chloroplast (Figure 4d) were observed in the cells of Au-NPs-treated plants during CA.

Nanopriming during CA led to an increase in chloroplast and peroxisomes sizes (Table 2). However, some reduction in the area of the mitochondria was observed (Table 2). During CA, nanopriming led to a decrease in the size of the grana but an increase in the size of the plastoglobuli and starch inclusions (Table 3). Moreover, nanopriming increased the number starch grains and decreased the number of plastoglobuli during CA (Table 3).

Additionally, nanopriming during CA did not affect photosynthesis, dark respiration, the relative expression level of *rbcS*, or the content of photosynthetic pigments (Table 1, Figure 2 and Figure 3); however, the relative expression of *rbcL* was higher in plants treated with Au-NPs (Figure 3).

The results of the experiments revealed that the contents of MDA and H_2_O_2_ in the leaves of Au-NPs-treated plants were higher than that of untreated plants during CA (Table 4). At this time, the SOD and CAT activities were greater in the plants treated with Au-NPs (Table 4). Activity of APX after cold exposure was decreased in plants treated with Au-NPs (Table 4). Au-NP treatment slightly decreased the content of soluble sugars in tomato leaves during CA (Table 1).

The cold tolerance of plants was increased during CA. Thus, tomato plants remained alive at 2 °C, and nanopriming reduced the electrolyte leakage by 7% (Table 5).

Analysis of the gold content in tomato tissues revealed that the seeds of plants after nanopriming contained gold. Moreover, trace amounts of gold were detected in the leaves of the plants grown from these seeds (Table 6).

In general, nanopriming caused an increase in cold tolerance; at the same time, NPs did not accumulate in the aboveground parts of the plants. Acceptable levels of heavy metals, such as lead, in plant tissues ranged from 0.5 to 10 mg/kg. These concentrations do not cause signs of toxicity in plants. The detection limits of gold in tissues (<0.05 mg/kg^−1^) using electrothermal atomic absorption spectroscopy suggest that it is not tactically contained in the shoots of treated plants.

The increase in cold tolerance was accompanied by a number of changes in plant metabolism. Some of these changes emerged under the influence of nanopriming under both control conditions and during CA. However, it is possible to identify specific changes that occurred under the influence of nanopriming only under control conditions or during CA (Figure 5).

Under control conditions, nanopriming enhanced photosynthesis rate, the relative expression of *rbcS* gen, the number of grana in chloroplast, and the content of photosynthetic pigments in leaves. During CA, nanopriming did not affect photosynthesis and photosynthetic pigments content but led to a decrease in the size of mitochondria and an increase in the area of peroxisomes in leaf cells. Regardless of the temperature, nanopriming increased the size of chloroplast, the relative expression of *rbcL* gene, the DW of tomato shoots, the content of MDA and H_2_O_2_, and the activity of CAT and SOD but decreased the activity of APX and sugar content in tomato leaves.

## 3. Discussion

### 3.1. Cold Tolerance

Under the action of abiotic factors, the leakage of electrolytes from tissues increases, which indicates a violation of the selective permeability of cell membranes. This may be associated with the disruption of the membrane structure or lipid complex. The permeability of cell membranes is an early indicator of changes in the physiological functions of a plant, so its change serves as a criterion for assessing the tolerance of plant tissues to abiotic stressors. This study revealed that nanopriming increased the cold tolerance of tomato both under control conditions and after CA (Table 5). Thus, under control conditions, nanopriming reduced the yield of electrolytes from tissues by 5%, whereas after CA, it reduced the yield of electrolytes from tissues by 10%. The decrease in electrolyte leakage indicates membrane stability and greater tolerance to LT.

### 3.2. Growth Parameters

Nanopriming did not affect the growth of tomatoes but increased the DW of their leaves regardless of temperature conditions (Table 1, Figure 5). This may be due to the accumulation of metabolites in leaf tissues. However, our experiments revealed that nanopriming did not increase the content of soluble sugars in leaves (Table 1). The increase in DW may be due to the accumulation of other primary or secondary metabolites, such as proteins, amino acids, or flavonoids. For example, Al_2_O_3_, TiO_2_, and NiO-NPs increased the accumulation of flavonoids in fennel [27], and Zn and Fe-NPs stimulated phenol accumulation in rose plants [28].

### 3.3. Photosynthetic Apparatus

Nanopriming in control conditions (at 22 °C) induced an increase in the photosynthetic rate (Figure 2), which is likely due to the stimulation of chlorophyll and carotenoid accumulation (Table 1) as well as an increase in the number of chloroplast grana and relative expression of *rbcL* and *rbcS* genes (Figure 3), which encode the large and small subunits of RuBisCo. Chlorophylls are part of the reaction centers and light harvesting complex, while carotenoids also have antioxidant functions, protecting chlorophylls and membrane lipids from photodegradation [29]. Similar changes under the influence of NPs were reported in other studies. For example, TiO_2_-NPs increase the rate of photosynthesis in tomatoes [30] and affect the expression of genes involved in the synthesis of RuBisCo [31]. Ag, TiO_2_, and ZnO-NPs increase the chlorophyll content in leaves [32,33,34,35,36]. In addition, the positive effects of NPs on PSA are often attributed to the plasmon resonance effect, i.e., their ability to increase the absorption of light by chlorophyll molecules [37,38] and quench excessive excitation by capturing the energy of excited electrons [38,39].

Nanopriming did not affect photosynthesis, the content of chlorophylls and carotenoids in leaves, or the relative expression of RuBisCo gene *rbcS* during CA. However, the expression of the *rbcL* gene encoding the large RuBisCo subunit and located in chloroplast DNA was higher in the plants treated with Au-NPs.

A number of specific changes in the ultrastructure of chloroplasts were observed in plants grown from AuNPs-treated seeds and then adapted to cold (Figure 5): an increase in the areas of plastoglobuli and starch inclusion area and a decrease in area of grana (Table 3), a shift in the membrane system to the periphery of the chloroplast (Figure 1d), the appearance of clusters of organelles (mitochondria and peroxisomes) near plastids, and invagination of the cytoplasm in chloroplasts (Figure 4). The decrease in grana size may reflect the redistribution of photosystems in the membranes. Photosystem II is known to be most sensitive to photoinhibition [40] and is localized in grana membranes, whereas the more stable photosystem I is located in the thylakoid membranes of the stroma [41]. The increase in the size of plastoglobuli may be related to reorganization of the chloroplast membrane system. This is most likely caused by the proteins and lipids that are produced during the reorganization of the thylakoid membrane system, which accumulate specifically in plastoglobuli, which grow in size [42,43]. Invaginations in chloroplasts are thought to promote enhanced transport of metabolites and signaling molecules between plastids and the cytoplasm [43]. The accumulation of mitochondria and peroxisomes near the chloroplast favors contact between them. Considering that LT slows all physiological processes and reactions in cells [44], we can most likely assume the adaptive significance of this structural feature.

Regardless of the temperature conditions, an increase in the size of chloroplasts was observed in the plants treated with Au-NPs. The increase in the chloroplast area under control conditions may be related to the accumulation of enzymes such as RuBisCo in their stroma or the increase in the size of starch grains. During CA, the increase in the chloroplast area may be related to the slowing of the outflow of assimilates from the chloroplast into the cytoplasm as well as to the inhibition of phloem flux under the influence of LT [43,45].

### 3.4. Respiration Rate and Mitochondria Size

This study revealed that nanopriming did not affect the respiration rate in control conditions or during CA (Figure 2 and Figure 5). LT led to an increase in the size of mitochondria, but it was less pronounced in the Au-NPs-treated plants (Table 2). An increase in the size of mitochondria may be a compensatory reaction aimed at maintaining reduced respiration and energy metabolism in LT conditions.

### 3.5. Pro-/Antioxidant Balance

It is thought that NPs have the ability to activate nonspecific plant adaptive pathways. These include AOS, the most important nonspecific protection system against oxidative stress accompanied by any unfavorable factor. The opinions of scientists on the effects of NPs on the pro-/antioxidant balance of plants are not clear. On the one hand, NPs are able to induce the development of oxidative stress by increasing ROS accumulation, and on the other hand, they can regulate this process by influencing AOS components. Our studies revealed that, regardless of temperature, nanopriming stimulates MDA and H_2_O_2_ accumulation and increases the activity of CAT and SOD (Table 4). These data indicate that NPs can act as triggers of oxidative stress. Other investigations reported similar results, such as reactive oxygen species (ROS) accumulation and increased AOS activity under the influence of different NPs [46,47,48]. One common reaction of a plant to the ingress of chemicals is the increased production of ROS and the subsequent development of oxidative stress under the impact of NPs [49]. When NPs come into contact with proteins, lipids, and other macromolecules, they can form highly active biochemical complexes. It is feasible for free radical intermediates involved in ROS production to develop within these complexes. The increase in oxidative stress caused by NPs can be explained by their high chemical activity [50]. However, an increase in the intensity of oxidative processes may also be a consequence of the increased intensity of photosynthesis and metabolism in general under NPs [50]. It is significant to note the signaling functions of ROS and lipid peroxidation (LPO) products. These functions are realized by controlling the expression of certain nuclear and chloroplast genes, transcription factors, hormone and redox signaling systems, and calcium status [51]. It is believed that there is a certain threshold concentration for plants, which becomes fatal; but all depends on the type of plant and its response to oxidative stress as well as the nature of the stressor itself. For example, CuO-NPs in low concentrations (10–50 mg L^−1^) stimulated the activity of AOS enzymes and in high concentrations (100–1000 mg L^−1^) led to the development of oxidative stress in rice [52].

Notably, the enzymatic components of AOS are important for plant survival under abiotic stress. For example, SOD catalyzes the reaction of O_2_^∙−^ dismutation to oxygen molecules and H_2_O_2_. SOD is the “first line” of protection for biologically important molecules (DNA, proteins, etc.) against ROS [53]. CAT catalyzes the dismutation of two H_2_O_2_ molecules to water and oxygen in peroxisomes, whereas peroxidases reduce H_2_O_2_ to water without releasing oxygen [54]. APX and CAT are enzymes that differ in terms of their affinity for H_2_O_2_ and their requirement for reducing power during H_2_O_2_ metabolism. APX efficiently eliminates even very low levels of H_2_O_2_ when ascorbate is used as its substrate. CAT degrades H_2_O_2_ without any reducing power and is mainly active at relatively high H_2_O_2_ concentrations [55]. Since the H_2_O_2_ content in the leaves increased in the Au-NPs-treated plants, the CAT activity was also high. A decrease in APX activity may be associated with a decrease in the content of ascorbic acid, which may work as an independent antioxidant. There are scientific references that APX activity can be suppressed with a decrease in ascorbate concentration [56]. High concentrations of H_2_O_2_ may also be the cause of the inhibition of APX activity. It must be emphasized that nanopriming increased POX activity in control conditions and decreased it during CA. Under control conditions, the content of H_2_O_2_ was increased almost twofold in plants treated with Au-NPs. This is likely due to the increased activity of POX, which helps to eliminate H_2_O_2_.

The increase in MDA content in tomato plants may be associated with an increase in the activity of enzymes that break down fatty acids (lipoxygenases) as well as enzymes that utilize MDA itself (aldehyde dehydrogenases) [57]. There is evidence in the literature that an increase in MDA may be not only a sign of membrane damage but also a sign of adaptation. For example, MDA may play a positive role by activating regulatory genes involved in plant defense under oxidative stress. According to the literature [58], moderate oxidative stress and oxidative signaling are essential for supporting plant growth and development. In particular, H_2_O_2_ produced by NADPH oxidases plays an important role in the response of tomato plants to oxidative stress as a signaling agent. These findings suggest that MDA is related to signaling and the regulation of key biological functions, such as meristem activity and flower opening [59].

Based on the above, we conclude that the increase in MDA and H_2_O_2_ under the influence of nanopriming is associated not so much with oxidative stress as with signaling activation.

### 3.6. Nanopriming as a Method to Increase Plant Stress Tolerance and Metabolism

Most likely, nanopriming can be considered a method to increase plant stress tolerance and a potential technology for advancing sustainable agriculture. On the one hand, nanopriming is a cost-effective method with low economic costs. In addition, it is a rather environmentally friendly strategy does not promote the accumulation of nanomaterials in living objects. Notably, the intensive use of NPs has raised concerns about their possible accumulation in ecosystems. Concerns are heightened in agriculture, where soils are deliberately exposed to products containing NPs, such as substances with pesticide or antibacterial activity, and irrigation with untreated wastewater also occurs [59,60,61,62,63]. Soil, which is one of the final sinks of NPs, can represent a source of NPs entering food webs. Therefore, developing strategies for the use of NPs in experimental biology and agriculture is highly relevant. In this direction, a number of important aspects should be taken into account: (1) the stability and chemical nature of the NPs, (2) the dose–effect, and (3) the need to test different objects. It is believed that NPs in low concentrations act as a trigger involving non-specific plant protection mechanisms. The size of NPs, their shape, and chemical nature are of great importance [50]. As a rule, aqueous colloidal solutions containing spherical metal NPs 10–50 nm in size have a positive effect on plants. Small (10 nm) NPs enter the tissues in excessive quantities, seeping through the pores of the cell. Large (50 nm) NPs can cause mechanical damage to cells [17,50]. The possibility of applying nanopriming in areas with different climates, soil conditions, and methods of plant protection from pests due to the need to increase not only stress resistance but also crop yields is also necessary.

## 4. Materials and Methods

### 4.1. Synthesis and Description of the Au-NPs

Gold nanospheres were produced via the citrate method [64]. A total of 250 mL of a 0.01% aqueous chloroauric acid solution was heated to 100 °C in an Erlenmeyer flask via a magnetic stirrer and a reflux water condenser. After adding 7.75 mL of 1% aqueous sodium citrate solution, the mixture was boiled for 30 min or until a red sol was produced. Freshly prepared aqueous colloidal Au-NPs solutions were transferred into sterile glass vials with tight-fitting lids and stored at 4 °C.

The diameter of the synthesized Au-NPs was determined via absorption spectroscopy, transmission electron microscopy, and dynamic light scattering methods (Figure 6). Measurements of the hydrodynamic radius and zeta potential of the NPs were performed on a Zetasizer Nano-ZS (Malvern, UK). Electron microscopic studies were carried out on a LIBRA 120 digital electron microscope (Carl Zeiss, Oberkochen, Germany). A UV-vis Specord 250 double-beam spectrophotometer (Analytik Jena, Jena, Germany) was used to measure the absorption and elastic scattering spectra. The maximum absorption wavelength of the obtained sol was λ_max_ = 518 nm, and the optical density was A_520_ = 1.18. The average diameter of the obtained NPs was 15.4 nm. The number of particles in 1 mL at A_520_ = 1 was 1.6 × 10^12^. The zeta potential was −19 mV. The maximum concentration of gold in solution was 57 µg mL^−1^. The stock solution of NPs was diluted with distilled water immediately before concentration tests and experiments.

### 4.2. Plant Material

Tomato seeds (*Solanum lycopersicum* L., Solanaceae) of the variety Kulon (Voronezh Vegetable Research Station, Voronezh, Russia) were used in the experiments. Seeds were soaked in Au-NPs solutions (20 µg mL^−1^) for 24 h, where 5 mL of Au-NPs solution was used for 100 seeds in each experiment variant. Afterwards, the seeds were transferred to a substrate based on Agrobalt-S peat (Rostorfinvest, Moscow, Russia) in climatic cameras at a temperature of 22 °C, a relative humidity of 60–70%, an illumination of 100 μmol m^−2^ s ^−1^, and a photoperiod of 16 h. As a control, we considered a variant where seeds were soaked in distilled water.

### 4.3. LT Exposure

Some of the plants aged 21 d were acclimated to LT in a KBW-240 climatic chamber (Binder, Tuttlingen, Germany) at 9 °C for 5 d, keeping other conditions unchanged (relative humidity of 60–70%, illumination of 100 μmol m^−2^ s^−1^, and a photoperiod of 16 h). The design of the experiment is shown in Figure 7.

### 4.4. Electrolyte Leakage

The plants were dislocated in a MIR-153 climatic chamber (Sanyo, Osaka, Japan) at 4 °C, 2 °C, and 0 °C for a period of 24 h. The electrical conductivity of the aqueous extracts was determined according to Hepburn et al. [65]. The following formula was used to determine electrolyte leakage from leaf tissues (V, %):V = 100 × (L_o_/L_k_),
where L_o_ is the electrical conductivity of the test sample before or after LT exposure, and L_k_ is the electrical conductivity of the same sample after boiling.

To determine electrolyte leakage in tomato leaves, seven biological and four statistical repetitions were used.

### 4.5. Growth Parameters

The height of the tomato shoots was recorded at 21 d of growth (before LT exposure) and 26 d (after LT exposure). Thirty plants were used to measure the height of the shoot. After drying at 100–105 °C to a constant weight, the leaf DW was determined as a percentage of the initial wet weight of the sample.

### 4.6. Ultrastructure of Leaf Mesophyll Cells

For electron microscopy, the samples from the middle parts of 3–4 leaves were fixed via the standard method for 4 h in 2.5% glutaraldehyde in 0.1 M phosphate buffer (pH 7.4) at 0–4 °C with postfixation in 2% OsO_4_. The samples were dehydrated in an ethanol series (20–100%), and the material was embedded in Epon-812. Ultrathin slices of leaves were prepared via an LKB-3 ultramicrotome (LKB, Sollentuna, Sweden). The slices were observed under a LIBRA120 electron microscope (ZEISS, Oberkohen, Germany). Morphometric analysis of the ultrastructures of the cells in the first subepidermal layer of the mesophyll was performed via the programs ITEM 5.0 and Axion Vision 4.8.

The ultrastructure was studied on 100 cells for each experiment variant.

### 4.7. CO_2_–Gas Exchange

CO_2_–gas exchange was measured in an open-type unit equipped with a URAS 2T IR gas analyzer (Hartmann und Braun, Heidenheim, Germany). The measurements of gas exchange included the determination of the net CO_2_ assimilation rates and dark respiration, both of which are expressed in mg CO_2_ g^−1^ DW per h [66].

CO_2_–gas exchange was determined in 20 plants of each variant.

### 4.8. Photosynthetic Pigment Contents

The amounts of chlorophyll a (Chl *a*), chlorophyll b (Chl *b*), and total carotenoids (Car) in the leaves were measured via a Genesys 10UV spectrophotometer (Thermo Electron Corporation, Waltham, MA, USA) at wavelengths that corresponded to the pigments’ absorption maxima in an 80% acetone solution—663, 646, and 470 nm, respectively. The pigment concentrations were calculated via the following formulas [67]:C*car* = (1000*D*_470_ − 3.27C*a* − 104C*b*)/198C*a* = 12.21*D*_663_ − 2.81*D*_646_C*b* = 20.3*D*_646_ − 5.03*D*_663_

The pigment content was expressed in mg g^−1^ DW.

To determine the contents of pigments, SIX biological repetitions were used.

### 4.9. Sugars Content

The content of fructose in the obtained extracts was determined according to Nakamura [68] by conversion of the sucrose content. The glucose content was determined via the glucose oxidase method via the Olvex Diagnosticum Kit (Vital Diagnostics, Moscow, Russia). Sugar content was defined as the sum of sucrose, fructose, and glucose contents expressed in mg g^−1^ leaf DW.

To determine the contents of sugars, six biological and three statistical repetitions were used.

### 4.10. RNA Isolation and qRT-PCR

Next, 100 mg of each leaf tissue sample was used to obtain total RNA by using the Extract RNA reagent (Evrogen, Moscow, Russia). Total RNA preparations were treated with DNase I (Thermo Fisher Scientific, Waltham, MA, USA) to remove residual genomic DNA impurities. cDNA was synthesized via an MMLV-RT Kit (Evrogen, Moscow, Russia).

Real-time quantitative PCR (RT-qPCR) was conducted on a CFX96 Touch™ instrument (Bio-Rad, Hercules, CA, USA) with SYBR Green I intercalating dye (Evrogen, Moscow, Russia). The 25 μL reaction mixture used for quantitative PCR contained 5 μL of qPCR mix HS SYBR (Evrogen, Moscow, Russia), 0.3 μmol of each primer, and 15 ng of the cDNA template. The amplification conditions were as follows: 95 °C for 5 min, followed by 40 cycles of 95 °C for 15 s, 60 °C for 30 s, and 72 °C for 30 s.

The primer-BLAST (https://www.ncbi.nlm.nih.gov/tools/primer-blast/ (accessed on 1 June 2025) and OligoAnalyzer™ tools (https://eu.idtdna.com/pages/tools/oligoanalyzer (accessed on 1 June 2025)) were used to select gene-specific primers. The sequences of the primer pairs used for real-time PCR are listed in Table 1. The transcript levels were normalized to the expression of the reference gene EF1 (elongation factor) for tomato. The Pfaffl technique [69] was used to determine the relative expression levels of the *rbcL* and *rbcS* genes, which are genes encoding the large and small subunits of RuBisCo.

To determine the level of relevant gene expression, three biological and three statistical repetitions were used.

### 4.11. Contents of MDA and H_2_O_2_

The LPO level was determined as the content of MDA. We used the thiobarbituric acid reaction by Heath and Packer [70] with minor modifications. The nonspecific absorbance of the supernatants at 600 nm was subtracted from the absorbance at 532 nm. The MDA equivalent was calculated on the basis of the resulting difference using the extinction coefficient of 155 mM cm^−1^.

The H_2_O_2_ content in 100 mg of leaf tissue was measured via a Peroxide Assay Kit (Sigma-Aldrich, St. Louis, MO, USA) according to the manufacturer’s instructions. The absorption of the solution was determined at 585 nm against a 3% H_2_O_2_ standard. The content of H_2_O_2_ was evaluated via a standard calibration curve constructed for hydrogen peroxide in acetone (from 0.1 to 1 mmol). The contents of MDA and H_2_O_2_ were calculated in μmol g^−1^ DW of leaves.

To determine the MDA and H_2_O_2_ contents, six biological repetitions were used.

### 4.12. Extraction of Soluble Proteins and Activity of Antioxidant Enzymes

Soluble proteins were isolated from 500 mg of leaf tissue in a buffer solution containing 50 mM Tris-HCl (pH 7.6), 3 mM EDTA, 250 mM sucrose, 3.6 mM cysteine, 5 mM ascorbic acid, 3 mM MgCl_2_, 2 mM DTT, and 2 mM phenylmethanesulfonyl fluoride. The extract was purified on a PD-10 midiTrap G-25 column (GE Healthcare, Chicago, IL, USA). A bicinchoninic kit (Sigma-Aldrich, St. Louis, MO, USA) was used to measure the amount of protein in the sample and to identify the activity of the antioxidant ferments.

The superoxide dismutase (SOD, EC 1.15.1.1) activity was determined via the method of [71], which is based on the generation of superoxide radicals in the course of riboflavin photooxidation, which is enhanced by an indicator trap (NBT, nitroblue tetrazolium). The reaction mixture contained 1.5% L-methionine, 0.14% NBT, and 1% Triton X-100 at a ratio of 3:1:0.75. Optical density measurements were performed at 560 nm. The inhibition of formazan production by 50% was recognized as a unit of SOD activity, which was converted to 1 g of protein.

Guaiacol peroxidase activity (POX, EC 1.11.1.7) was determined according to the methods of Kumar and Knowles [72], with modifications. The method is based on the oxidation of guaiacol to tetraguaiacol. The reaction mixture contained 0.3 mL of 0.15 mM guaiacol, 0.9 mL of Tris-HCl buffer (10 mM, pH 7.6), and 0.3 mL of the supernatant. Before optical density measurement, 0.3 mL of 0.1 mM H_2_O_2_ was added to the cuvette. The optical density was measured at 470 nm. The reaction proceeded at 25 °C for 1 min at intervals of 10 s. POX activity was expressed as μmol guaiacol g^−1^ protein × min.

The activity of ascorbate peroxidase (APX, EC 1.11.1.11) was determined according to Nakano and Asada [73] with modifications. The method is based on the measurement of the rate of ascorbate decrease in the reaction catalyzed by ascorbate peroxidase. The reaction mixture included 0.3 mL of protein extract, 100 μL of 5 mmol ascorbate, 50 μL of 0.1 mmol EDTA, and 2.75 mL of Tris-HCl buffer (10 mM, pH 7.6). Before optical density measurement, 100 μL of 0.1 mmol H_2_O_2_ was added to the cuvette. Optical density was measured at 290 nm for 1 min at intervals of 10 s. APX activity was expressed as μmol ascorbate g^−1^ protein × min.

The catalase activity (CAT, EC 1.11.1.6) was measured by the rate of the H_2_O_2_ decomposition reaction [73]. The final reaction consisted of 2.8 mL of Tris-HCl buffer, pH 7.6; 0.3 mL of enzyme extract; and 0.3 mL of H_2_O_2_ (3 mmol) for 3 min. Observation of the depletion of H_2_O_2_ in accordance with the decrease in absorbance at 240 nm was used to measure the CAT activity in the supernatant. CAT activity was expressed as μmol H_2_O_2_ g^−1^ protein × min.

Enzyme activity was determined in fresh material. The volume of the buffer used for the extraction of enzymes was 1 mL.

To determine the AOS activity, six biological and three statistical repetitions were used.

### 4.13. Content of Gold in Plant Tissues

Next, 100 mg of plant tissues or seeds dried at 60 °C were placed into fine glass tubes and mineralized at 120 °C with 10 mL of aqua regia (3:1 (*v*/*v*) 12 M hydrochloric acid:15.6 M concentrated nitric acid) until a clear and colorless solution was obtained. The digest solutions were diluted with 0.1 M HCl and analyzed via electrothermal atomic absorption spectroscopy (ETAAS) via a Shimadzu AA-7000 (Shimadzu, Kyoto, Japan) with a hollow-cathode lamp (Photron, Warren, Australia) and a pyrolytic-coated graphite cuvette (Shimadzu, Kyoto, Japan). A gold linear calibration curve over the range of 0 to 20 ng mL^−1^ was prepared from 1000 mg L^−1^ standard gold solution in hydrochloric acid. Parallel sample blank solutions were used to calculate the gold concentration background.

To determine the gold content in leaves, 10 biological repetitions were used.

### 4.14. Statistical Analysis

Statistical significance was calculated via one-way ANOVA with Tukey’s test (*p* < 0.05) via Origin 7.0 software. The tables and figures show the mean values and their standard deviations.

## 5. Conclusions

In this study, it was found for the first time that seed priming with Au-NPs leads to the formation of tomato plants with altered PSA and pro-/antioxidant status. As a result, the tolerance of plants to cold significantly increases. At the same time, nanoparticles do not accumulate in the aboveground parts of the plants. Some of physiological and structural changes emerged under the influence of nanopriming under both control conditions and during CA. However, it is possible to identify specific changes that occurred under the influence of nanopriming only under control conditions or during CA. This study highlights the potential of the use of Au-NPs as a promising strategy to mitigate cold stress in tomato plants by affecting photosynthesis and the antioxidant system.

## Figures and Tables

**Figure 1 plants-15-00083-f001:**
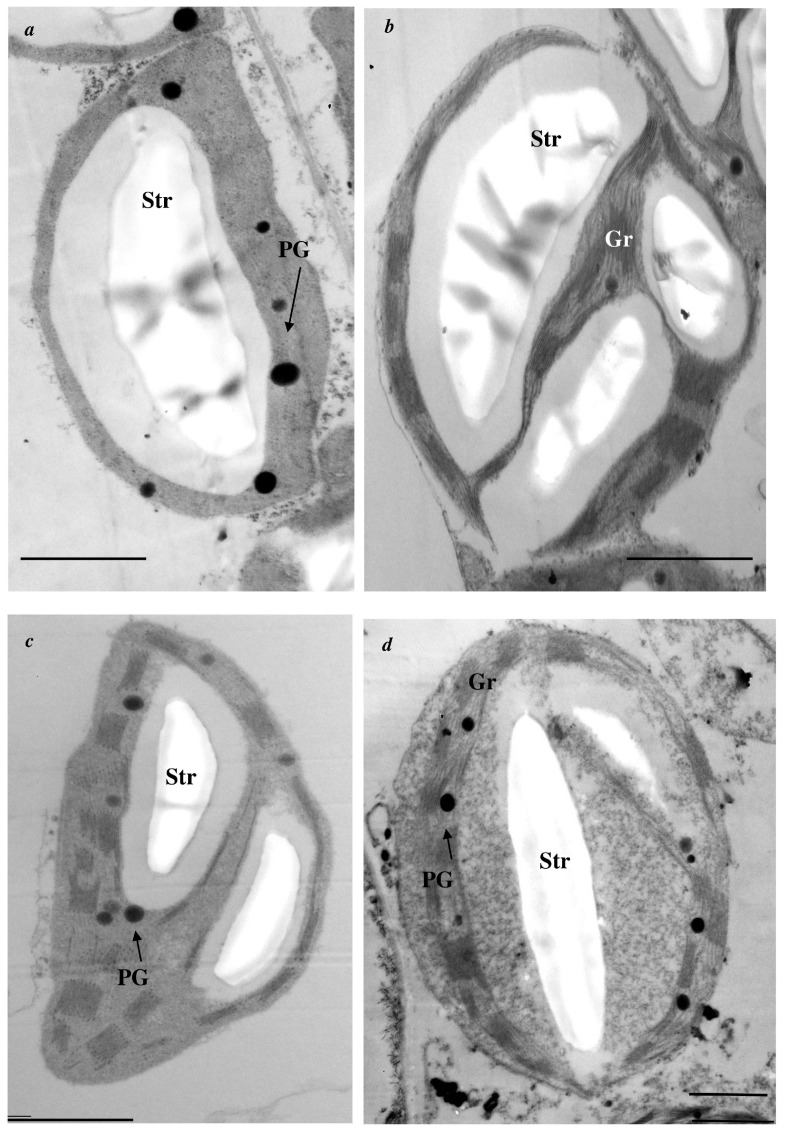
Effect of nanopriming on the ultrastructure of chloroplasts in tomato (*Solanum lycopersicum*) leaves at 22 °C (**a**,**b**) and after (**c**,**d**) cold adaptation (9 °C, 5 d): (**a**,**c**)—control; (**b**,**d**)—nanopriming. Gr—grana; PG—plastoglobuli; Str—starch inclusion. Bar: 1 µm.

**Figure 2 plants-15-00083-f002:**
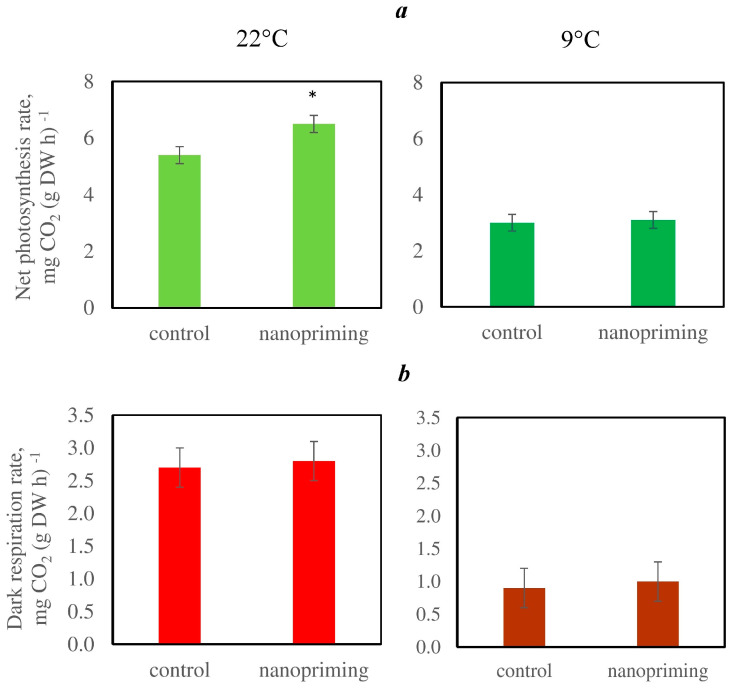
Effects of nanopriming on the net photosynthesis (**a**) and dark respiration (**b**) rates in tomato (*Solanum lycopersicum*) leaves at 22 °C and after cold adaptation (9 °C for 5 d). The figure shows the mean values and their standard deviations. The means that significantly differ from control at *p* < 0.05 are denoted by asterisk.

**Figure 3 plants-15-00083-f003:**
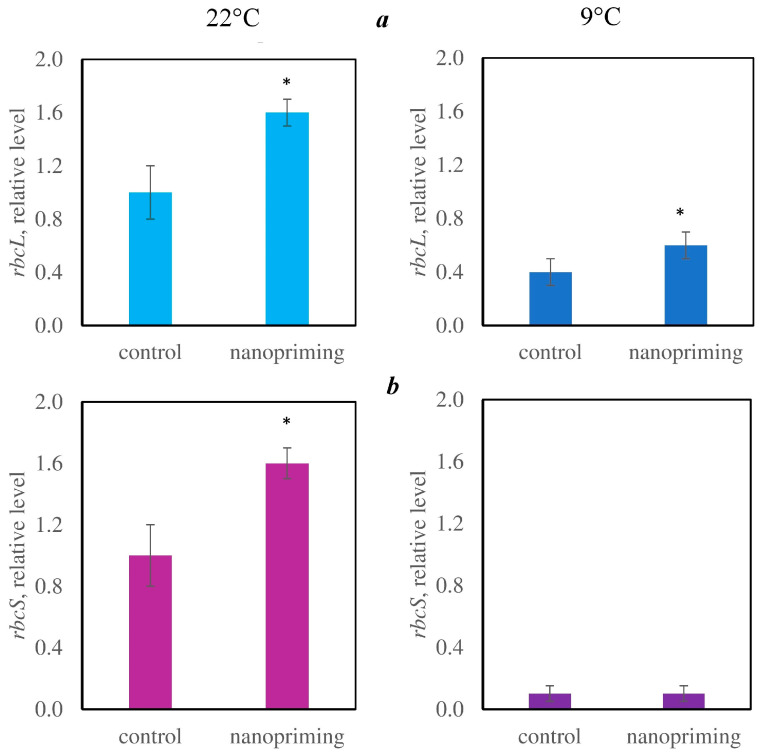
Effects of nanopriming on the relative expression levels of the genes *rbcL* (**a**) and *rbcS* (**b**) in tomato (*Solanum lycopersicum)* leaves at 22 °C and after cold adaptation (9 °C for 5 d). The figure shows the mean values and their standard deviations. The means that significantly differ from control at *p* < 0.05 are denoted by asterisk.

**Figure 4 plants-15-00083-f004:**
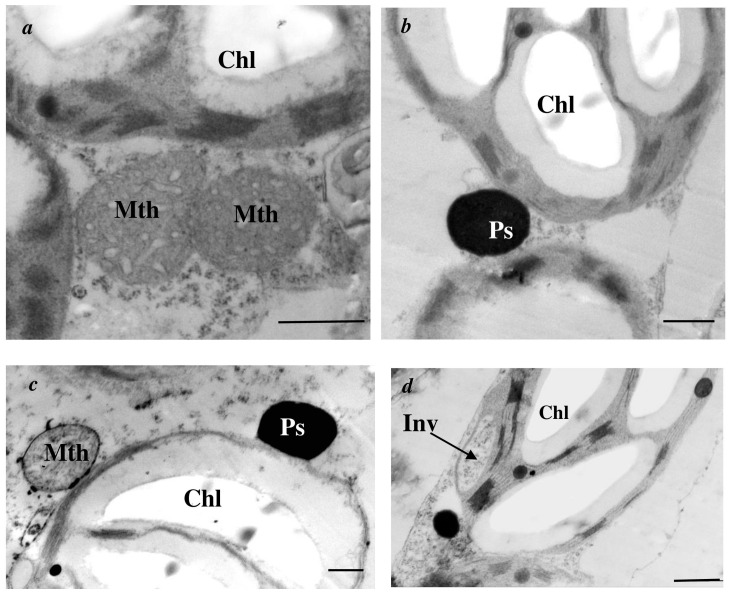
Effect of nanopriming on the ultrastructure of mesophyll cells in tomato (*Solanum lycopersicum*) leaves after cold adaptation (9 °C, 5 d): clusters of organelles (**a**,**c**), dark peroxisomes (**b**,**c**), invaginations of the cytoplasm within the chloroplast (**d**). Chl—chloroplast; Mth—mitochondria; Ps—peroxisome; Inv—invagination of the cytoplasm in the chloroplast. Bar: 0.5 µm.

**Figure 5 plants-15-00083-f005:**
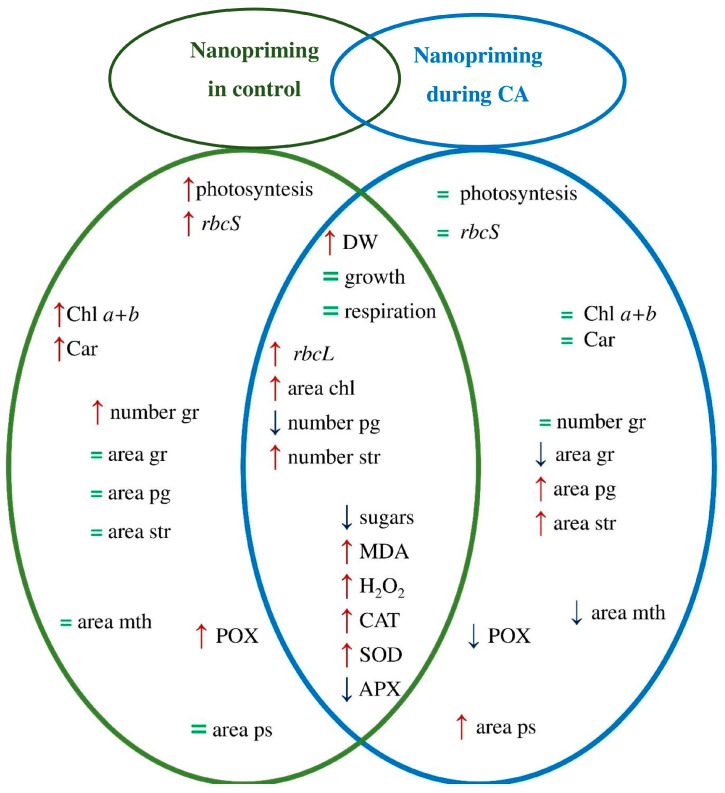
Effects of nanopriming on tomato (*Solanum lycopersicum*) in control conditions and during cold adaptation (CA): chl—chloroplast; gr—grana; pg—plastoglobuli; str—starch in chloroplast; mth—mitochondria; ps—peroxisome. Analysis of changes compared with the control variant: ↑—parameter increases; ↓—parameter decreases; =—no change in parameter.

**Figure 6 plants-15-00083-f006:**
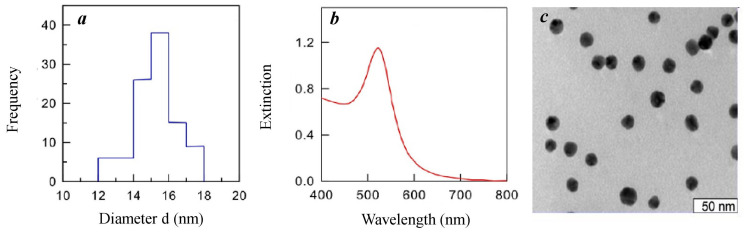
Size distribution (**a**), absorption spectrum (**b**), and transmission electron microscopy image (**c**) of gold nanoparticles with an average diameter of 15 nm.

**Figure 7 plants-15-00083-f007:**
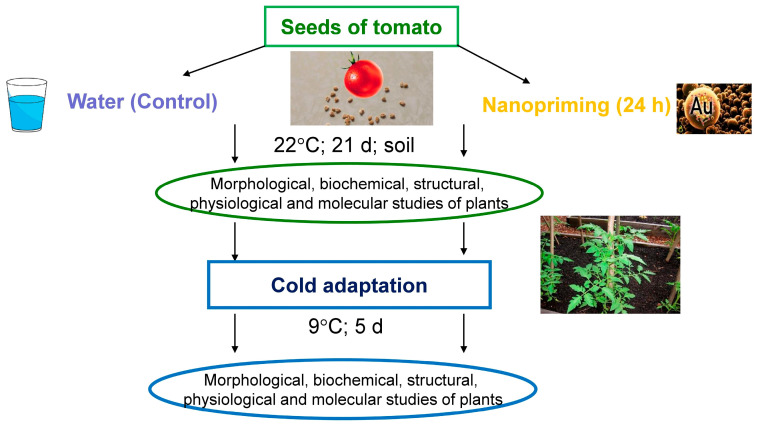
Experimental design.

**Table 1 plants-15-00083-t001:** Effects of nanopriming on the growth and contents of photosynthetic pigments and soluble sugars in tomato (*Solanum lycopersicum*) leaves at 22 °C and after cold adaptation (9 °C for 5 d).

Parameters	At 22 °C		At 9 °C	
	Control	Nanopriming	Control	Nanopriming
Shoot height, cm	6.97 ± 0.51	7.18 ± 0.48	7.70 ± 0.78	7.88 ± 0.66
Dry weight of shoot, g	0.15 ± 0.01	0.16 ± 0.01 *	0.16 ± 0.01	0.17 ± 0.01 *
Chl *a + b*, mg g^−1^ DW	8.17 ± 0.31	10.37 ± 0.65 *	6.70 ± 0.56	6.78 ± 0.26
Car, mg g^−1^ DW	1.37 ± 0.10	1.68 ± 0.11 *	1.44 ± 0.24	1.49 ± 0.05
Glu, mg g^−1^ DW	8.79 ± 0.05	6.04 ± 0.22 *	30.03 ± 0.34	21.98 ± 0.92 *
Fru, mg g^−1^ DW	14.60 ± 0.96	6.26 ± 0.55 *	78.02 ± 2.12	63.06 ± 0.93 *
Suc, mg g^−1^ DW	14.13 ± 1.35	12.49 ± 0.39	50.74 ± 1.96	38.62 ± 4.09 *
Sum of sugars, mg g^−1^ DW	37.51 ± 0.41	24.71 ± 1.13 *	158.69 ± 3.67	124.65 ± 5.86 *

The table shows the mean values and their standard deviations. The means that significantly differ from control at *p* < 0.05 are denoted by asterisk. Chl *a + b*—sum of pigments chlorophyll *a + b*; Car—content of carotenoids; Glu—content of glucose; Fru—content of fructose; Suc—content of sucrose; DW—dry weight.

**Table 2 plants-15-00083-t002:** Effect of nanopriming on the sizes of organelles in tomato (*Solanum lycopersicum)* leaves at 22 °C and after cold adaptation (9 °C for 5 d).

Parameters	At 22 °C		At 9 °C	
	Control	Nanopriming	Control	Nanopriming
Chloroplast area, µm^2^	7.92 ± 1.66	9.52 ± 1.50 *	6.08 ± 1.10	8.68 ± 1.10 *
Mitochondria area, µm^2^	0.23 ± 0.09	0.22 ± 0.09	0.46 ± 0.08	0.33 ± 0.09 *
Peroxisome area, µm^2^	0.30 ± 0.12	0.26 ± 0.07	0.26 ± 0.07	0.49 ± 0.12 *

The table shows the mean values and their standard deviations. The means that significantly differ from control at *p* < 0.05 are denoted by asterisk.

**Table 3 plants-15-00083-t003:** Effect of nanopriming on chloroplast ultrastructure in tomato leaves at 22 °C and after cold adaptation (9 °C for 5 d).

Parameters	At 22 °C		At 9 °C	
	Control	Nanopriming	Control	Nanopriming
	Number, pcs ^1^		Number, pcs ^1^	
Grana	9.50 ± 3.10	16.70 ± 5.00 *	27.44 ± 5.10	22.71 ± 5.10
Plastoglobuli	4.77 ± 3.25	3.16 ± 1.70 *	6.30 ± 2.05	3.91 ± 1.80 *
Starch inclusions	2.06 ± 0.51	2.95 ± 0.61 *	1.73 ± 0.70	2.85 ± 0.76 *
	Area, µm^2^		Area, µm^2^	
Grana	0.05 ± 0.01	0.05 ± 0.01	0.05 ± 0.01	0.04 ± 0.01 *
Plastoglobuli	0.02 ± 0.01	0.02 ± 0.01	0.02 ± 0.01	0.03 ± 0.01 *
Starch inclusion	2.12 ± 1.17	2.48 ± 1.00	1.1 ± 0.41	2.70 ± 1.10 *

^1^ The number of structural elements (grana and plastoglobuli) was counted per 10 µm^2^ of chloroplast area (due to the variation in the chloroplast area in each variant of the experiment). The table shows the mean values and their standard deviations. The means that significantly differ from control at *p* < 0.05 are denoted by asterisk.

**Table 4 plants-15-00083-t004:** Effects of nanopriming on the pro-/antioxidant parameters of tomato (*Solanum lycopersicum*) leaves at 22 °C and after cold adaptation (9 °C for 5 d).

Parameters	At 22 °C		At 9 °C	
	Control	Nanopriming	Control	Nanopriming
APX, µmol g^−1^ protein	190.67 ± 14.02	49.39 ± 4.00 *	91.82 ± 7.99	53.27 ± 3.51 *
POX, µmol g^−1^ protein	39.13 ± 3.00	96.77 ± 7.51 *	217.13 ± 16.01	169.58 ± 10.51 *
CAT, µmol H_2_O_2_ g^−1^ protein	0.33 ± 0.11	2.99 ± 0.10 *	0.49 ± 0.09	0.68 ± 0.122 *
SOD, a.u. g^−1^ protein	0.88 ± 0.11	1.69 ± 0.11 *	1.41 ± 0.08	2.20 ± 0.13 *
MDA, µM g^−1^ DW	14.50 ± 0.85	28.38 ± 1.05 *	24.63 ± 1.34	30.97 ± 0.99 *
H_2_O_2_, µM g^−1^ DW	21.37 ± 0.80	38.43 ± 1.02 *	11.56 ± 0.37	13.29 ± 0.30 *

The table shows the mean values and their standard deviations. The means that significantly differ from control at *p* < 0.05 are denoted by asterisk. APX—ascorbate peroxidase; POX—guaiacol peroxidase; CAT—catalase; SOD—superoxide dismutase; MDA—malondialdehyde.

**Table 5 plants-15-00083-t005:** Effect of nanopriming on electrolyte leakage (%) in tomato (*Solanum lycopersicum)* leaves at 22 °C and after cold adaptation (9 °C for 5 d).

Temperature Treatment	At 22 °C		At 9 °C	
	Control	Nanopriming	Control	Nanopriming
4 °C	17.14 ± 4.10	12.23 ± 2.79 *	21.23 ± 3.69	11.73 ± 2.79 *
2 °C	88.09 ± 1.52	90.33 ± 5.41	20.19 ± 2.55	13.08 ± 3.12 *
0 °C	87.91 ± 2.90	84.56 ± 1.90	78.38 ± 23.96	69.33 ± 24.32

The table shows the mean values and their standard deviations. The means that significantly differ from control at *p* < 0.05 are denoted by asterisk.

**Table 6 plants-15-00083-t006:** The content of gold in the seeds and leaves of tomato (*Solanum lycopersicum*) plants after nanopriming.

Tomato Organs	Content of Gold (mg kg^−1^ DW)	
	Control	Nanopriming
Seeds	<0.05	14.45 ± 0.10
Leaves	<0.05	0.51 ± 0.01

The table shows the mean values and their standard deviations.

## Data Availability

The datasets used and/or analyzed during the current study are available from the corresponding author upon reasonable request.

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
