# Peer review of "Plants2026, 15(1), 83;https://doi.org/10.3390/plants15010083"

_plants, 2025, doi:10.3390/plants15010083_

Round 1

Reviewer 1 Report (New Reviewer)

Comments and Suggestions for Authors

Plants-3974434: The manuscript presents an interesting and well-structured study exploring the effects of Au-NP seed priming on cold tolerance and metabolism in tomato plants. The topic is timely and relevant, addressing sustainable nanotechnology applications in plant physiology. However, several sections require clarification, improved methodological transparency, and stronger discussion of the mechanisms and broader implications. Detailed comments are provided below. Minor grammatical issues appear throughout (e.g., “do not affect” should be “did not affect,” L108; “in process of CA” should be “during CA,” L126). A careful proofreading or English language edit is recommended.

L37: Typo: “action o LT” → “action of LT.”

L62–74: The rationale for using Au-NPs should be strengthened. Clarify why Au-NPs were selected over other NPs.

L83: “Colloidal Au-NPs solution” should be “colloidal Au-NP solution.”

L87: Add a clear hypothesis statement at the end of the introduction.

L123-130: Clarify whether chloroplast ultrastructure was quantified in multiple cells per plant.

Tables 2-4, 6: Consider reporting mean ± SE to two decimal places for consistency.

Figures 3-4: Require higher resolution.

L207–217: The authors report elevated H₂O₂ and MDA levels in Au-NP-treated plants. However, whether these values represent stress or signaling activation should be discussed.

L239–246: The Au content results are interesting but would benefit from discussion on the detection limits (<0.05 mg kg⁻¹) and analytical sensitivity.

Table 7 is very dense. Please consider moving to supplementary materials and referring to key trends in the text.

Lines 314–328: The explanation that NPs act as ROS triggers is valid but should be balanced with evidence showing signaling versus toxicity. Include at least one citation discussing threshold concentrations for beneficial vs. harmful ROS levels.

Lines 330–369: Excellent mechanistic interpretation of ultrastructural changes. However, please provide higher-level synthesis rather than repeating all observed trends.

Lines 394–411: The section on antioxidant enzymes is informative, but the causal linkage between APX inhibition and ascorbate depletion is speculative. Please consider softening the statement.

L451–456: Specify the number of seeds per treatment group and whether randomization was applied during germination and growth.

L462: I believe the “variant” here meant “treatment”.

L464: How did you determine the concentration and treatment time length of Au-NPs (20 μg mL-1 24 h)?

L459–467: Clarify if cold adaptation was continuous (5 days) or intermittent, and confirm whether the temperature change was gradual or abrupt.

L502–507: Specify environmental parameters (light intensity, CO₂ concentration) during gas exchange measurements.

L551–561: Provide the full chemical formula for the thiobarbituric acid (TBA) reaction and whether results were normalized to fresh or dry weight.

L616–618: The statistical methods lack justification of sample size or power analysis. Include discussion or reference supporting replication adequacy.

L621: Make the Conclusions concise. Detailed information should be moved to Discussion section.

Comments on the Quality of English Language

Minor grammatical issues appear throughout (e.g., “do not affect” should be “did not affect,” L108; “in process of CA” should be “during CA,” L126). A careful proofreading or English language edit is recommended.

Author Response

Dear Reviewer,

Thank you very much for your encouraging feedback about our work. The responses to your comments are summarized in the attached file and highlighted in red in the manuscript.

Reviewer 2 Report (Previous Reviewer 2)

Comments and Suggestions for Authors

The reviewed manuscript is a resubmission of a previously evaluated version. Given this fact, the current content of the manuscript should be assessed in the context of the comments raised in the earlier review rounds. In my opinion, the revisions introduced by the authors are not sufficient to consider the manuscript suitable for publication, as comments provided in review process of previous submission were not editorial or interpretative in nature but concern the core experimental methodology, which directly affects the validity of the study’s conclusions.

However, in order to give the authors an opportunity to address all the issues raised in the previous reviews, I ask the authors to provide point-by-point response explaining how these concerns have now been resolved. Below is a summary of the most important issues concerning the current version of the manuscript.

1

As indicated in the previous review rounds, the description provided by the authors in Section 4.3 remains imprecise. In lines 462–468, the authors indicated that the experiment included four treatment variants:

“Four variants of the experiment were used (Figure 7):

1 – control – plants grown under control conditions from seeds soaked in distilled water;

2 – nanopriming – plants grown from seeds soaked with Au-NPs (20 μg mL⁻¹, 24 h) under control conditions;

3 – CA – plants grown under control conditions (from seeds soaked in distilled water) and then adapted at 9°C for 5 days;

4 – nanopriming + CA – plants grown from seeds treated with Au-NPs (20 μg mL⁻¹, 24 h) under control conditions and then adapted at 9°C for 5 days.”

However, in lines 486–487, the authors further indicate that: “The height of the tomato shoots was recorded at 21 days of growth (before LT exposure) and 26 days (after LT exposure).

This means that until day 21, only two treatment variants were present — control and nanopriming. The other two variants, CA and nanopriming + CA, appeared only later, once the plants were exposed to cold stress. Comparing the values of parameters resulted from application of four treatment variants is methodologically incorrect, because the data were collected at different time points — on day 21 for the first two variants and on day 26 for the latter two.

The experiment is two-factor experiment, where one factor is the type of treatment and the other is exposure to cold stress. Such an experimental design determines what the treatment variants should be included. Moreover, values of parameters tested should be measured at the same time point. Only then obtained results can be compared.

1a.

In their accompanying e-mail, the authors provided a list of publications in which, according to them, a similar experimental setup was used. I kindly ask the authors to indicate the specific sections or lines in each of those papers where details of the experimental design are described, so that this claim can be verified.

2.

As for the general comment related to the Results section, the authors continue to report standard errors instead of standard deviations. In my opinion, this approach is not appropriate, as the key aspect in this type of experiment is the variability among the plants subjected to each treatment variant, not the variability of repeated measurements within a single plant (the latter being what the standard error primarily represents). Taking the example of the data on shoot height (expressed in centimetres), a mean value of 7 cm accompanied by an error value of only 0.1 does not reflect the variability among all plants within a given treatment variant. Rather, it indicates that the measurements for individual plants were taken precisely. Therefore, standard deviation would be a more suitable measure to describe the biological variation observed in this experiment.

3.

The method of selecting the plants used for subsequent measurements remains unclear to me. The authors state that a total of 30 plants were used for measuring shoot height, and that, depending on the parameter analyzed, various number of plants (biological repetitions) were then selected for further biochemical or physiological analyses (e.g. lines 548-549 - "To determine the level of relevant gene expression 3 biological and 3 statistical repetitions were used"; lines 527-528 - "To determine the contents of sugars 6 biological and 3 statistical repetitions were 527 used"). In the context of selecting plants from the total of 30 individuals used in the experiment, and considering the different numbers of plants (biological replicates) analyzed for various parameters, I understand that these plants were randomly selected. However, this approach still appears methodologically incorrect, as it negatively affects the representativeness of the samples. A consistent sampling method should be applied across all measured parameters to ensure that the obtained data accurately reflect the variability within each treatment group.

Author Response

Dear Reviewer,

We have read your recommendations and comments again and are sincerely grateful to you. Your comments allowed us to review our work, see the flaws in it and fix them. We agree with your comments and have completely changed the manuscript, trying to focus on your views.

We are attaching responses to your comments in a separate file and thank you for your help.

Round 2

Reviewer 1 Report (New Reviewer)

Comments and Suggestions for Authors

I am satisfied with the changes made by the authors. I would support the publication in its current form.

Comments on the Quality of English Language

Minor grammatical issues appear throughout (e.g., “do not affect” should be “did not affect,” L108; “in process of CA” should be “during CA,” L126). A careful proofreading or English language edit is recommended.

Author Response

Dear Reviewer,

We sincerely thank you for your time and attention to our research. We have carefully checked the English language of the manuscript again, guided by your advice.

We are always happy to cooperate.

With best regards,

Yuliya Venzhik and team

Reviewer 2 Report (Previous Reviewer 2)

Comments and Suggestions for Authors

Comments and suggestions are provided in the attached file.

Author Response

Dear Reviewer,

We would like to express our sincere gratitude for your valuable input and attention towards our research. We deeply regret that you may not be satisfied with our work at this time. We appreciate your comments, as they allow us to better understand our concerns and address them in a more effective manner.

In response to your feedback, we have prepared the following points.

  1. The fact that the authors implemented this change only at the editorial level does not contribute to an actual improvement in the scientific value of the manuscript. I understand that the authors have now separated the measurement points by temperature (I would additionally recommend indicating also the day of growth, to clearly indicate the time passed between both measurement terms). However, the crucial question remains: what clear conclusions can be drawn from this experiment? I continue to believe that the experimental design is fundamentally flawed. Plants naturally grow and develop over time, and therefore, the values of measured parameters are expected to change in subsequent growth days. In this context, I refer specifically to the fact that measurements were performed at different time points: day 21 of growth for plants maintained at 22 °C, and day 26 for plants kept at 9 °C. Consequently, the observed differences reflect not only the effect of temperature, but also the effect of time. Taking shoot weight as an example, one might incorrectly conclude that lower temperature resulted in higher shoot weight. This interpretation would be misleading, as the obtained values result from two overlapping effects—time of plant growth and temperature conditions. Under such experimental design, it is not possible to determine which of these two factors contributed more strongly to the obtained result. This limitation could be resolved if additional variants of treatment were included— i.e., if a set of plants remained under optimal temperature conditions and were measured simultaneously with those shifted to cold conditions. Only such a comparative approach would allow for a reliable separation of the effects of time and temperature, and thus enable a valid interpretation of treatment impact. Regardless of how the authors restructure the table or analyze the results, it is impossible to unambiguously determine to what extent the observed differences are due to treatment effects and to what extent they simply reflect time-dependent growth.

Response. Our main task is to study the effect of nanopriming on plants in control conditions and during cold adaptation. We used plants of the same age when comparing the control variant and nanopriming. We thank the distinguished reviewer for his comments. We excluded passages that talk about the effects of low temperatures on plants from text of manuscript.

An increase in the dry weight of shoots is not a mistake, but is associated with the accumulation of sugars, which are formed during adaptation to cold as protective substances.

  1. I have examined two of the references indicated by the authors. My comments on these papers are provided below. If the authors believe that their assumption is indeed supported by indicated research, I kindly ask them to indicate publications that report two-factor experiments (i.e., with an experimental design comparable to the one used in the present study), paste fragment of these papers and to explain explicitly how these results substantiate their claims. Fu X, Feng Y, Zhang Y, Bi H, Ai X. Salicylic acid improves chilling tolerance via CsNPR1–CsICE1 interaction in grafted cucumbers. Horticulture research. 2024 Oct;11(10):uhae231 https://doi.org/10.1093/hr/uhae231 Page 11, left column, paragraph 2 Chilling and SA treatments All experiments were conducted in an artificial climate chamber to investigate the impact of grafting on cucumber cold tolerance. Two types of cucumber plants were used: the self-rooted (cucumber as rootstock and scion, Cs/Cs) and grafted cucumber plants (with pumpkin as rootstock, Cs/Cm). These plants were treated at 8◦C/5◦C for 24, 72, and 120 h, with those treated at 25◦C/18◦C serving as controls. The chilling injury index (CI), reactive oxygen (ROS) accumulation, and cold-responsive genes expression were measured at 72 h. In my view, the experiment presented in the cited work should be classified as a single-factor design, in which temperature was the only variable under investigation. The comparisons were made exclusively between plants grown under different temperature conditions, which indicates that no second independent factor was introduced or controlled in the experimental setup. Ghosh PK, Sultana S, Keya SS, Nihad SA, Shams SN, Hossain MS, Tahiat T, Rahman MA, Rahman MM, Raza A. Ethanol-mediated cold stress tolerance in sorghum seedlings through photosynthetic adaptation, antioxidant defense, and osmoprotectant enhancement. Plant Stress. 2024 Mar 1;11:100401. https://doi.org/10.1016/j.stress.2024.100401 Page 2, right column, paragraph 1 Overall, the experiment included four treatments: (i) water-pretreated control seedlings (Ctrl), (ii) ethanol-pretreated seedlings (Eth), (iii) water-pretreated cold-stressed seedlings (Cold), and (iv) ethanol-pretreated cold-stressed seedlings (Eth+Cold). In my view, this experiment should be classified as a two-factor design, with treatment as the first factor and temperature as the second factor. Accordingly, the authors tested four treatment variants. Overall, in my opinion, the authors’ assumption that the proposed experimental design is appropriate has not been sufficiently supported by evidence from other publications.

Response. We would like to clarify that there may be various experimental designs and perspectives on conducting an experiment. In this study, we did not consider the impact of temperature on plants but rather primarily focused on studying the effect of nanoparticles under various conditions. It appears to us that the design we have chosen is valid and has a right to exist. We appreciate the Reviewer's perspective and respect his opinion. We intend to take into consideration all the recommendations given by the reviewer in our future endeavors. The opinion of experts with extensive experience in experimental biology holds great significance for us, and we remain open to listening and adapting for the better. It is crucial to consider diverse perspectives on this matter, and we greatly appreciate your comments.

We present several studies with a similar experimental scheme:

1) H. Hasanpour, R. Maali_Amiri, and H. Zeinali. Effect of TiO2 Nanoparticles on Metabolic Limitations to Photosynthesis Under Cold in Chickpea Russian Journal of Plant Physiology, 2015, Vol. 62, No. 6, pp. 779–787.DOI: 10.1134/S1021443715060096

2) Rahmat Mohammadi & Reza Maali-Amiri & Alireza Abbasi. Effect of TiO2 Nanoparticles on Chickpea Response to Cold Stress. Biol Trace Elem Res (2013) 152:403–410. DOI: 10.1007/s12011-013-9631-x

3) R. Mohammadi, R. Maali_Amiri, and N. L. Mantri Effect of TiO2 Nanoparticles on Oxidative Damage and Antioxidant Defense Systems in Chickpea Seedlings during Cold Stress. Russian Journal of Plant Physiology, 2014, Vol. 61, No. 6, pp. 768–775. DOI: 10.1134/S1021443714050124

  1. In my opinion, the dataset presented by the authors still contains inconsistencies. Below I provide a screen of tables for comparison between the previous version of the manuscript (where standard errors were reported) and the current version (where standard deviations are presented). The intention of this comparison is to evaluate whether the values are internally consistent. It seems unlikely that the standard deviation for root dry weight of the 30 examined plants per each variant of treatment (section 4.5.) is equal only to 0.01 for all columns (measurement terms and variants of treatment) Such uniformity across columns is unexpected, particularly considering that the standard deviation reported for shoot height is substantially higher.

Response. The low values of the standard deviation relative to this indicator are explained by the fact that we measured the dry weight in 1 g of the fresh mass of the shoots. The figures presented show how many grams of dry matter are contained in 1 gram of the fresh mass of shoots. Since we used this methodical approach, the average error and standard deviation were small for each experimental variant.

  1. I appreciate the authors’ willingness to improve methodological consistency in future experiments, however this does not fully address the core concern raised in reviewer’s comment. The photo attached indeed suggests general visual uniformity of the plants, but it does not replace a methodological explanation nor does it confirm biological homogeneity with respect to the measured traits.

Response. We regret that we were not able to meet the expectations of the Reviewer. We appreciate the valuable comments you have provided, and we believe that most of them are valid. Working with you has been a valuable experience for us, as it has allowed us to refine our manuscript and approach our data in a more informed manner.

Thank you for your assistance.

Best regards,

Yuliya Venzhik and team.

This manuscript is a resubmission of an earlier submission. The following is a list of the peer review reports and author responses from that submission.

Round 1

Reviewer 1 Report

Comments and Suggestions for Authors

This is an interesting study, contributing to a better understanding of the effect of seed priming with gold nanoparticles on the cold tolerance of tomato plants. The significance and novelty of the study are clearly described. The manuscript is well written, and the results are properly analyzed and discussed. I have some minor remarks:

Why did you use such a low light intensity (100 µmol m-2 s-1) to grow tomato plants?

What was the light intensity for CO2 assimilation measurements?

Line 150 – The results on the effect of nanopriming on mitochondria area should be better described. It is written that ”Reduction of mitochondria area was observed only in the variant nanopriming+LT”. Compared to what? The results presented in Table 3 show that nanopriming  under control conditions did not affect mitochondrial area compared to the control. Mitochondria area increased as a result of LT and nanopriming+LT, but to a lesser extent in the latter.

Line 203 – It is better to use “their content” instead of “these processes”

Please correct the sentence in line 221, there is some repetition there.

Line 223 – The results presented in Table 5 show that the activity of APX decreased in plants from nanopriming variant (49.4) compared to the control (190.7).

Author Response

For Reviewer1

Dear Reviewer,

We thank you for your attention to our work and its high appreciation. Your questions and comments help us to better understand our material and improve the manuscript. We have made the changes proposed by the reviewer (marked in red in the manuscript). Answers to the comments of the Reviewer are summarized below.

Reviewer #1:

This is an interesting study, contributing to a better understanding of the effect of seed priming with gold nanoparticles on the cold tolerance of tomato plants. The significance and novelty of the study are clearly described. The manuscript is well written, and the results are properly analyzed and discussed. I have some minor remarks:

  1. Why did you use such a low light intensity (100 µmol m-2s-1) to grow tomato plants? What was the light intensity for CO2 assimilation measurements?

Response. The tomato plants were grown under conditions of a climate chamber, where the optimal lighting for the plants is 100-150 µmol m-2 s-1. We did not initially use intense light because during the experiments the plants were cooled and we measured their tolerance to low temperatures. We relied on the fact that in low temperature conditions light is always excessive and this can lead to photoinhibition. All experiments were done under the same lighting conditions.

  1. Line 150 – The results on the effect of nanopriming on mitochondria area should be better described. It is written that ”Reduction of mitochondria area was observed only in the variant nanopriming+LT”. Compared to what? The results presented in Table 3 show that nanopriming  under control conditions did not affect mitochondrial area compared to the control. Mitochondria area increased as a result of LT and nanopriming+LT, but to a lesser extent in the latter.

Response. We thank the reviewer for his attention - his perfect comment accurately reflects the essence of what is happening. Changes were made in the manuscript (lines 145-147 ).

  1. Line 203 – It is better to use “their content” instead of “these processes”.

Response. Changes were made in the manuscript.

  1. Please correct the sentence in line 221, there is some repetition there.

Response. Changes were made in the manuscript.

  1. Line 223 – The results presented in Table 5 show that the activity of APX decreased in plants from nanopriming variant (49.4) compared to the control (190.7).

Response. We thank the reviewer for his attention - changes were made in the manuscript (lines 207-209).

We express our gratitude to the esteemed Reviewer once more for his consideration of our manuscript,

Best wishes,

Yuliya Venzhik and coauthors.

Reviewer 2 Report

Comments and Suggestions for Authors

The authors addressed the issue of increasing plant tolerance to abiotic stress. Given the challenges of modern agriculture and the unfavourable growing conditions, this issue is of high importance. In lines 410-411, the authors indicate that: "This study establishes for the first time that nanopriming with Au-NPs can increase cold tolerance in tomato plants".

In the context of the results presented in Tables 1-6 and Figures 3 and 4, I ask the authors to indicate the number of samples analyzed for each parameter. In lines 618-619, the authors indicate that: “the biochemical experiments were performed in 3-4 biological and 6-7 statistical repetitions. Growth was measured on 30 plants in each experiment variant”. However, such a description seems to be not sufficiently precise. Moreover, I would ask the authors to provide the raw data for the "growth" related results that are presented in Table 2. The standard deviations for the shoot length and dry weight of shoot parameters appear very low considering that 30 plants were measured.

As for the results presented in Table 2, did the authors also measure the fresh weight of the shoot? The authors point to significant differences in the dry weight of the shoot. However, the differences may result from differences related to shoot length parameters meaning that a higher dry weight may simply result from a longer stem. Providing the date for the percentage of dry weight content would be better. Moreover, table 2 is not self-explained. I ask the authors to provide the meaning of abbreviations used in the table (e.g. Chl a+b, Car, Glu)

As for the discussion, in line 261, the authors indicated that: “The study showed that nanopriming causes an increase in cold tolerance of tomato 261 both in control conditions and after LT (Table 1)”. Table 1 presents the results for the effect of tested treatment variants o on electrolyte leakage in leaves. In my opinion, the scientific relationship between electrolyte leakage and cold stress tolerance is not adequately addressed in the article. For a non-expert reviewer, it might not be obvious why this parameter is used to describe the cold tolerance of plants.

In lines 266-267, the authors refer to Figure 5 in the sentence "However, it is possible to identify specific changes that occurred under the influence of nanopriming only in control or LT conditions." If I understand correctly, the figure summarizes the results presented in this paper, meaning it indicates whether the values for a given parameter increased, decreased, or remained unchanged, Firstly, the figure description would need to be more precise. For example, for Chl a+b, the figure description at the bottom should indicate the unit for this parameter or indicate whether it refers to a quantity or some other parameter describing it. Secondly, the figure clearly summarizes the results, but I wonder if the discussion section is a for such a figure. If it is intended to summarize the results, it can be presented in the results section. If it is to be placed in the discussion section, it seems that it is necessary to discuss it with the summary of other scientific papers.

As for the general comment for the discussion section and the whole paper, I ask the authors for an explanation regarding the following issue. I understand that this work presents for the first time the results obtained for the use of Au-NPs. However, in the discussion section (e.g. lines 412-422,) the authors refer to NPs as a group of compounds, not to a specific type of NP. Is it that way that the entire group of nanoparticles (meaning nanoparticles based on different elements, not only Au-NPs) has the same or similar mode of action or does the mode of action depend on the element of the nanoparticle e.g. Au-NPs). Moreover, how do the characteristics of a nanoparticle e.g. its size, influence the action of the nanoparticle?

In line 469 there is an indication of 20g/ml and in line 481 the is an indication of 20 μg/mL used. Which one is the correct one?

As for section 4.2, what was the volume of solution in which the seeds were soaked?

In section 4.5. growth parameters, the authors indicated that: “The height of the tomato shoots was recorded at 21 d of growth (before LT exposition) 498 and 26 d (after LT exposition)”. Are the results from measurement terms included in the paper?

Author Response

For Reviewer2

Dear Reviewer,

We appreciate your insightful feedback. Your recommendations helped us to revisit and improve the material. We have made every effort to update the manuscript in response to your suggestions (highlighted in red in the manuscript). Responses to the reviewers' comments are summarized below.

Reviewer #2:

The authors addressed the issue of increasing plant tolerance to abiotic stress. Given the challenges of modern agriculture and the unfavourable growing conditions, this issue is of high importance. In lines 410-411, the authors indicate that: "This study establishes for the first time that nanopriming with Au-NPs can increase cold tolerance in tomato plants".

  1. In the context of the results presented in Tables 1-6 and Figures 3 and 4, I ask the authors to indicate the number of samples analyzed for each parameter. In lines 618-619, the authors indicate that: “the biochemical experiments were performed in 3-4 biological and 6-7 statistical repetitions. Growth was measured on 30 plants in each experimentvariant”. However, such a description seems to be not sufficiently precise. Moreover, I would ask the authors to provide the raw data for the "growth" related results that are presented in Table 2. The standard deviations for the shoot length and dry weight of shoot parameters appear very low considering that 30 plants were measured.

Response. We thank the esteemed Reviewer for his attention and interest. This is a very correct remark - statistical analysis is very important. Therefore, we have expanded the description of our data for statistical analysis in the section Methodology (line 594-603) and show it here. To determine the electrolyte leakage in tomato leaves 4 biological and 7 statistical repetitions were used; to determine the dry weight of shoots, content of pigments and sugars, MDA and H2O2 content and AOS activity - 3 biological and 6 statistical repetitions were used;  to determine the level of relevant gene expression - 3 biological and 3 statistical repetitions were used. The length of the shoots was determined on 30 plants in each variant, CO2-gas exchange - on 20 plants in each variant.  To determine the gold content in leaves 10 biological and 3 statistical repetitions were used. We used 100 micro photos in each variant for analysis of ultrastructure parameters and 20 plants in each variant - for detection of CO2-gas exchange. We present the requested source data in a supplementary file. The tables and figures show the mean values and their standard errors (not standard deviations).

  1. As for the results presented in Table 2, did the authors also measure the fresh weight of the shoot? The authors point to significant differences in the dry weight of the shoot. However, the differences may result from differences related to shoot length parameters meaning that a higher dry weight may simply result from a longer stem. Providing the date for the percentage of dry weight content would be better. Moreover, table 2 is not self-explained. I ask the authors to provide the meaning of abbreviations used in the table (e.g. Chl a+b, Car, Glu).

Response. We haven’t measured the fresh weight of tomato leaf (shoot), but we have data on the percentage of dry weight in 1 gram of fresh weight. We submitted this data in the supplementary file. If the Reviewer makes such a decision, we can present these data in the manuscript. We have provided relevant comments to table 2.

  1. As for the discussion, in line 261, the authors indicated that: “The study showed that nanopriming causes an increase in cold tolerance of tomato 261 both in control conditions and after LT (Table 1)”. Table 1 presents the results for the effect of tested treatment variants o on electrolyte leakage in leaves. In my opinion, the scientific relationship between electrolyte leakage and cold stress tolerance is not adequately addressed in the article. For a non-expert reviewer, it might not be obvious why this parameter is used to describe the cold tolerance of plants.

Response. We thank the Reviewer for the comment, the relevant remarks are included in the Discussion (link 268-278).

  1. In lines 266-267, the authors refer to Figure 5 in the sentence "However, it is possible to identify specific changes that occurred under the influence of nanopriming only in control or LT conditions." If I understand correctly, the figure summarizes the results presented in this paper, meaning it indicates whether the values for a given parameter increased, decreased, or remained unchanged, Firstly, the figure description would need to be more precise. For example, for Chl a+b, the figure description at the bottom should indicate the unit for this parameter or indicate whether it refers to a quantity or some other parameter describing it. Secondly, the figure clearly summarizes the results, but I wonder if the discussion section is a for such a figure. If it is intended to summarize the results, it can be presented in the results section. If it is to be placed in the discussion section, it seems that it is necessary to discuss it with the summary of other scientific papers.

Response. We thank the Reviewer for this comment, the relevant remarks are added in the Results.

  1. As for the general comment for the discussion section and the whole paper, I ask the authors for an explanation regarding the following issue. I understand that this work presents for the first time the results obtained for the use of Au-NPs. However, in the discussion section (e.g. lines 412-422,) the authors refer to NPs as a group of compounds, not to a specific type of NP. Is it that way that the entire group of nanoparticles (meaning nanoparticles based on different elements, not only Au-NPs) has the same or similar mode of action or does the mode of action depend on the element of the nanoparticle e.g. Au-NPs). Moreover, how do the characteristics of a nanoparticle e.g. its size, influence the action of the nanoparticle?

Response. It is believed that nanoparticles in low concentrations act approximately the same way, including non-specific plant protection mechanisms. However, this issue is being actively discussed and so far, there is no exact answer to it. The size of nanoparticles, their shape and chemical nature are also of great importance. As a rule, aqueous colloidal solutions containing spherical metal nanoparticles with a size of 10-50 nm have a positive effect on plants. Small (˂ 10 nm) nanoparticles enter the tissue in excessive quantities, leaking through the pores of the cell. Large (˃ 50 nm) nanoparticles can cause mechanical damage to cells. (Sarraf, M.; Vishwakarma, K.; Kumar, V.; Arif, N.; Das, S.; Johnson, R.; Janeeshma, E.; Puthur, J.T.; Aliniaeifard, S.; Chauhan, D.K.; Fujita, M.  Metal/metalloid-based nanomaterials for plant abiotic stress tolerance: an overview of the mechanisms. Plants 2022, 11, 316. https://doi.org/10.3390/plants11030316; Pissuwan, D.; Poomrattanangoon, R.; Chungchaiyart, P. Trends in using gold nanoparticles for inducing cell differentiation: a review. ACS Appl. Nano Mater. 2022, 5, 3110. https://doi.org/10.1021/acsanm.1c04173).

  1. In line 469 there is an indication of 20g/ml and in line 481 the is an indication of 20 μg/mL used. Which one is the correct one?

Response. Thanks for the comment, corrections have been made, correct version - 20 μg/mL.

  1. As for section 4.2, what was the volume of solution in which the seeds were soaked?

Response. For 100 seeds 5 ml of solution was used.

  1. In section 4.5. growth parameters, the authors indicated that: “The height of the tomato shoots was recorded at 21 d of growth (before LT exposition) 498 and 26 d (after LT exposition)”. Are the results from measurement terms included in the paper?

Response. Day 21 - measurements before LT exposure (control and nanopriming variants), Day 26 - plants after exposure to LW for 5 days (i.e., these are the LW and LW + nano-priming variants).

We express our gratitude to the esteemed Reviewer once more for his consideration of our manuscript,

Best wishes,

Yuliya Venzhik and coauthors.

Reviewer 3 Report

Comments and Suggestions for Authors

Dear Authors,

I have read your manuscript carefully. Overall, I believe the work requires significant revision, particularly in terms of language clarity and expression. A thorough English language editing by a native speaker or a professional editor is strongly recommended to improve the readability and academic tone of the paper.

Moreover, I would appreciate a more extensive and in-depth analysis in the Discussion section, as this part currently lacks the depth needed to properly interpret the findings.

Below, I provide both general suggestions and specific line-by-line comments:

  • Please consider converting the unit notation from “20 µg/mL” to “20 µg mL⁻¹” consistently throughout the text and figures.

  • Avoid using the first-person plural ("we") in scientific writing.

  • Indicate the volumes of the solutions used in experimental procedures (e.g., volume of buffer used for enzyme extraction).

  • Avoid using superscripts to indicate statistical differences between treatments.

  • Table titles and captions should not be separated from the tables they refer to.

Line Comment
Abstract The abstract lacks clarity and does not adequately describe the experimental design. A brief description of how cold tolerance was assessed and treatments would help the reader better understand the scope of the study.
14 Consider rephrasing to: “Changes in metabolism contribute to enhanced cold tolerance.”
19 “It is…” – Should this be in past tense? Consider “It was…”
33 The word “risky” may not be appropriate in this context; suggest alternative terminology.
34 The phrase “at tached lifestyle” is unclear and possibly incorrect. Please rephrase.
38–39 Consider improving clarity and flow of the sentence.
49 Replace “There is evidence” with a more specific and formal alternative.
50–51 The meaning of this part is unclear – please clarify.
76 “Quite exciting” is informal; consider a more scientific phrasing.
84–85 Rephrase “The object of our study is tomato” to something like: “The study was conducted on tomato (Solanum lycopersicum)…”.
100 “Variants of experiment:” – do you mean “treatments”? If this refers to materials and methods, it is not necessary to repeat it in the figure caption. Also, explain all abbreviations used.
114 Inconsistency between “shoot height” in the text and “shoot length” in the table – please standardize.
149 “Morphometric analysis confirmed our results” – this would be more appropriate in the Discussion section.
214 Replace “The values” with “The means” if referring to average values.
226 “In the process of cold exposition” – do you mean during the cold treatment? Please clarify.
264 Consider rephrasing for clarity and precision.
306 Missing reference? Please verify.
316, 322, 327 Improve phrasing and clarity in all mentions of “Nanopriming”.
352 This part should be moved to the Results section and described accordingly.
410–493 These paragraphs are not directly related to the core findings. They could be significantly condensed and moved to the Conclusion if retained.
456  “λmax   A520
569 Specify whether enzyme activity was measured on fresh tissue.
598 Ensure consistent notation: “H₂O₂” instead of “H2O2”.

The topic is interesting and potentially relevant, but the manuscript requires considerable improvements in language, structure, and depth of analysis to meet publication standards. I encourage the authors to revise the manuscript thoroughly in accordance with the above comments.

Author Response

For Reviewer #3

Dear Reviewer,

We sincerely thank you for your attention to our work, as well as for your important comments and corrections. We note, that your comments allowed us to improve the manuscript considerably (corrections are colored in red in the manuscript). Responses to your comments are summarized below.

Reviewer #3:

I have read your manuscript carefully. Overall, I believe the work requires significant revision, particularly in terms of language clarity and expression. A thorough English language editing by a native speaker or a professional editor is strongly recommended to improve the readability and academic tone of the paper.

Moreover, I would appreciate a more extensive and in-depth analysis in the Discussion section, as this part currently lacks the depth needed to properly interpret the findings.

Below, I provide both general suggestions and specific line-by-line comments:

  • Please consider converting the unit notation from “20 µg/mL” to “20 µg mL⁻¹” consistently throughout the text and figures.
  • Avoid using the first-person plural ("we") in scientific writing.
  • Indicate the volumes of the solutions used in experimental procedures (e.g., volume of buffer used for enzyme extraction).
  • Avoid using superscripts to indicate statistical differences between treatments.
  • Table titles and captions should not be separated from the tables they refer to.

Response. We included corrections in manuscript in accordance with your recommendations. Thank you very much for your attention to our study.

Line

Comment

Abstract

Responce

The abstract lacks clarity and does not adequately describe the experimental design. A brief description of how cold tolerance was assessed and treatments would help the reader better understand the scope of the study.

Changes were included in the manuscript (lines 13-16).

14

Responce

Consider rephrasing to: “Changes in metabolism contribute to enhanced cold tolerance.”

Changes were included in the manuscript.

19

Responce

“It is…” – Should this be in past tense? Consider “It was…”

Changes were included in the manuscript.

33

Responce

The word “risky” may not be appropriate in this context; suggest alternative terminology.

Changes were included in the manuscript (lines 34-35 ).

34

Responce

The phrase “attached lifestyle” is unclear and possibly incorrect. Please rephrase.

We removed this sentence from the manuscript.

38–39

Responce

Consider improving clarity and flow of the sentence.

Changes were included in the manuscript.

49

Responce

Replace “There is evidence” with a more specific and formal alternative.

Changes were included in the manuscript (lines 49-52).

50–51

Responce

The meaning of this part is unclear – please clarify.

Changes were included in the manuscript.

76

Responce

“Quite exciting” is informal; consider a more scientific phrasing.

Changes were included in the manuscript (lines  75-76).

84–85

Responce

Rephrase “The object of our study is tomato” to something like: “The study was conducted on tomato (Solanum lycopersicum)…”.

Changes were included in the manuscript (lines 84-86).

100

Response

“Variants of experiment:” – do you mean “treatments”? If this refers to materials and methods, it is not necessary to repeat it in the figure caption. Also, explain all abbreviations used.

Changes were included in the manuscript.

114

Response

Inconsistency between “shoot height” in the text and “shoot length” in the table – please standardize.

Changes were included in the manuscript.

149

Response

“Morphometric analysis confirmed our results” – this would be more appropriate in the Discussion section.

We removed this sentence from the manuscript.

214

Response

Replace “The values” with “The means” if referring to average values.

Changes were included in the manuscript.

226

Response

“In the process of cold exposition” – do you mean during the cold treatment? Please clarify.

Changes were included in the manuscript.

264

Response

Consider rephrasing for clarity and precision.

Changes were included in the manuscript.

306

Response

Missing reference? Please verify.

Changes were included in the manuscript (line  312).

316, 322, 327

Response

Improve phrasing and clarity in all mentions of “Nanopriming”.

Changes were included in the manuscript (line 322, 328,332).

352

Response

This part should be moved to the Results section and described accordingly.

We ask the Reviewer to allow us to keep this table in this section. It seems to us that this table is more appropriate for Discussing. However, if the Reviewer insists, we are ready to revise the manuscript in the indicated direction.

410–493

Response

These paragraphs are not directly related to the core findings. They could be significantly condensed and moved to the Conclusion if retained.

Changes were included in the manuscript (line 605-622).

456

Response

 “λmax   A520

Changes were included in the manuscript (line  430).

569

Response

Specify whether enzyme activity was measured on fresh tissue.

Yes, enzyme activity was measured on fresh tissue.

598

Response

Ensure consistent notation: “H₂O₂” instead of “H2O2”.

Changes were included in the manuscript.

The topic is interesting and potentially relevant, but the manuscript requires considerable improvements in language, structure, and depth of analysis to meet publication standards. I encourage the authors to revise the manuscript thoroughly in accordance with the above comments.

We express our gratitude to the esteemed Reviewer once more for his consideration of our manuscript,

Best wishes,

Yuliya Venzhik and coauthors.

Reviewer 4 Report

Comments and Suggestions for Authors

In this scientific Article, the authors report on their studies regarding the effects of nanopriming with nanoparticles on the photosynthetic apparatus, redox state and cold tolerance of gold in tomato (Solanum lycopersicum L). Their study falls within the field of research on priming to improve cultivation in terms of crop resistance and yield, a sector that is therefore also very important in economic terms.
Considering the evolution of the various versions of the article, it is clear that the Authors clearly outline their research activity and the aims of the study in the introduction, which is sufficiently informative with up-to-date bibliographical references.
The experimental design is adequate, as are the techniques used to conduct the study, and these are reported in a simple and correct manner in the materials and methods section. The results are reported with figures and tables that are useful for understanding them, especially since some are described in an overly schematic manner in the text.
The data collected are interesting: changes in the photosynthetic apparatus, observations on mitochondria and peroxisomes, and the study of the effects of oxidative stress, such as malondialdehyde production and changes in the specific activity of the enzymes responsible for counteracting it, provide an informative picture of how treatment with nanoparticles affects the state of the plant. The discussion is good, although tables with results should be included in the results section and not in this section of the article. It is clear that the authors are well versed in this subject and, based on the results available to them, they reach conclusions that can be shared, also highlighting the critical issues inherent in the use of nanopriming in agriculture. In light of the conclusions, I wonder how the authors intend to continue these studies.
Some notes for the authors:
In the abstract, LT should be defined when first mentioned (line 16).
In the caption of Figure 5, the alignment of the arrows should be checked.
In Table 3, μkm2? Please check this unit of measurement.

Comments on the Quality of English Language

English language needs improvements

Round 2

Reviewer 2 Report

Comments and Suggestions for Authors

Comments on the authors' responses to the reviews are included in the attached file.

Author Response

The answers to the Reviewer are provided in the attached file.

Reviewer 3 Report

Comments and Suggestions for Authors

Dear authors

Below you can find some more remarks.

Line 37 - that destroys crops of important agricultural plants every year [1]  

Line 75 - biological sceinces

Line 76 - Research on these compounds as compounds 
Line 84 - which is an important agricultural cold-sensitive crop
Line 86 - We studied

Line 94 - We evaluated electrolyte leakage  - Write in passive voice

Figure 3: Convert the unit notation from ex “µg/mL” to “µg mL⁻¹” consistently throughout the text and figures.

In tables e.g. 2,  Correct the parameters column: Chl a+b, mg/g DW  to  Chl a+b(mg DW-1

Line 295-control conditions Describe them..

Line 306... the injection of a foreign chemical - Better writing 

Author Response

(The authors gave the same response as above.)

Round 3

Reviewer 2 Report

Comments and Suggestions for Authors

Comments on the authors' responses to the reviews are included in the attached file.

Author Response

Comments in file.

Round 4

Reviewer 2 Report

Comments and Suggestions for Authors The responses provided by the authors do not alter my overall recommendation regarding the paper.